# Drifting Streaming Peaks-Over-Threshold-Enhanced Self-Evolving Neural Networks for Short-Term Wind Farm Generation Forecast

Yunchuan Liu [1], Amir Ghasemkhani [2] and Lei Yang [3,*]

1 Division of Science Mathematics and Technology, Governors State University, University Park, IL 60484, USA
2 Department of Computer Science and Engineering, California State University San Bernardino, San Bernardino, CA 92407, USA
3 Department of Computer Science and Engineering, University of Nevada, Reno, Reno, NV 89557, USA
* Correspondence: leiy@unr.edu

**Abstract:** This paper investigates the short-term wind farm generation forecast. It is observed from the real wind farm generation measurements that wind farm generation exhibits distinct features, such as the non-stationarity and the heterogeneous dynamics of ramp and non-ramp events across different classes of wind turbines. To account for the distinct features of wind farm generation, we propose a Drifting Streaming Peaks-over-Threshold (DSPOT)-enhanced self-evolving neural networks-based short-term wind farm generation forecast. Using DSPOT, the proposed method first classifies the wind farm generation data into ramp and non-ramp datasets, where time-varying dynamics are taken into account by utilizing dynamic ramp thresholds to separate the ramp and non-ramp events. We then train different neural networks based on each dataset to learn the different dynamics of wind farm generation by the NeuroEvolution of Augmenting Topologies (NEAT), which can obtain the best network topology and weighting parameters. As the efficacy of the neural networks relies on the quality of the training datasets (i.e., the classification accuracy of the ramp and non-ramp events), a Bayesian optimization-based approach is developed to optimize the parameters of DSPOT to enhance the quality of the training datasets and the corresponding performance of the neural networks. Based on the developed self-evolving neural networks, both distributional and point forecasts are developed. The experimental results show that compared with other forecast approaches, the proposed forecast approach can substantially improve the forecast accuracy, especially for ramp events. The experiment results indicate that the accuracy improvement in a 60 min horizon forecast in terms of the mean absolute error (MAE) is at least 33.6% for the whole year data and at least 37% for the ramp events. Moreover, the distributional forecast in terms of the continuous rank probability score (CRPS) is improved by at least 35.8% for the whole year data and at least 35.2% for the ramp events.

**Keywords:** ramp events; short-term wind power forecast; distributional forecast; point forecast; Bayesian optimization; self-evolving neural networks

## 1. Introduction

To reduce the environmental impacts of the electricity system, much progress can be found to integrate renewable energy resources, such as solar and wind. Indeed, a substantial percentage of this renewable integration [1] comes from wind energy. Large-scale wind power integration has aroused new challenges in power system operations, particularly during wind power ramps. Large ramps have a significant influence on system economics and reliability. For instance, the unexpected wind power ramp events that occurred in Texas [2] caused a significant economic loss, and such cases were also reported in many other countries [3].

In this paper, we aim to develop accurate forecast approaches for a short-term wind power forecast that accounts for wind power ramps.

There are many studies on short-term wind power forecast using time-series models (e.g., the autoregressive model [4], autoregressive moving average model [5], Gaussian process (GP) [6], Kalman filtering (KF) [7], and Markov chains [8]). However, these studies cannot effectively capture the non-stationarity and the heterogeneous dynamics of wind farm generation. To address the problem of non-stationary wind generation, the empirical mode decomposition (EMD), complementary empirical mode decomposition (CEEMD) [9], improved complete ensemble empirical mode decomposition (iCEEMDAN) [10], hybrid model of LSTM and variational mode decomposition (VMD) [11,12], and ensemble empirical mode decomposition (EEMD)-based hybrid methods [13] are proposed, which use Intrinsic Mode Functions (IMFs) as a pre-processing measure and the product of the decomposition components as input for the prediction. However, finding an appropriate number of components or modes is challenging. Recently, artificial intelligence (AI)-based approaches were employed to many applications with success (e.g., Computer Vision (CV) [14], Natural Language Processing (NLP) [15], and Chess Playing [16]). Different neural network (NN)-based frameworks [11,17–34] were proposed for the wind generation forecast, e.g., artificial neural networks (ANN) [17], a wavelet neural network (WNN) [19], an adaptive neuro-fuzzy neural network (ANFIS) [18], the long short-term memory (LSTM) model [25], a convolutional neural network (CNN) [21,22,26], radial neural networks [23], a fuzzy wavelet neural network [24], a deep echo state network [20], a genetic LSTM [27], a K-shape- and K-means-guided deep convolutional recurrent network [28], a dynamic elastic NET (DELNET) [29], an attention temporal convolutional network (ATCN) [30], a spatio-temporal correlation model (STCM) based on convolutional neural networks long short-term memory (CNN-LSTM) [32], the extended deep sequence-to-sequence long short-term memory regression (STSR-LSTM) [33], a hybrid model with attention mechanism and complete ensemble empirical mode decomposition (CEEMDAN) [34], etc.

Although neural network (NN)-based methods may enhance the forecast accuracy to a certain degree, the existing NN-based approaches may have poor performance during ramp events, simply because the ramp and non-ramp events are not separated when training the NNs. It has been shown that NNs may perform poorly if extreme (or ramp) events are overlooked [35]. Previous studies [36,37] have revealed (1) the non-stationary and seasonal dynamics of wind farm generation and (2) the heterogeneous dynamics of non-ramp and ramp events. Moreover, as different classes of wind turbines are deployed in wind farms, we observe that the dynamics of the wind generation of different classes of wind turbines can be different (see Section 2). Thus, employing NNs without considering these distinct features of wind farm generation means wind farm generation cannot be accurately forecast, especially for ramp events (see Section 4). In the previous work, seasonal self-evolving neural networks [38] are built for different seasons and ramps are defined using fixed thresholds. However, it is observed that the dynamics of wind ramps may change within each season, and due to the time-varying dynamics of wind ramps, it is challenging to use fixed thresholds to accurately capture the dynamics of the wind ramps. To address this challenge, this paper proposes a dynamic threshold-based approach that can adapt to the time-varying dynamics of wind ramps.

Specifically, we propose Drifting Streaming Peaks-over-Threshold (DSPOT)-enhanced self-evolving neural networks that account for the time-varying dynamics of different wind turbines' power outputs during non-ramp and ramp events in order to achieve a better wind farm generation prediction. First, the proposed DSPOT approach leverages dynamic ramp thresholds to classify the wind generation data of each class of wind turbines into ramp and non-ramp datasets, which can account for the time-varying dynamics of the ramp and non-ramp events across different classes of wind turbines. Then, different NNs are trained for each dataset to learn the heterogeneous dynamics of the different classes of wind turbines' generation, in which the NeuroEvolution of Augmenting Topologies [39] is adopted to evolve the NNs in order to obtain the best network topology and weighting parameters. As the efficacy of NNs depends on the quality of the training datasets (i.e., the classification accuracy of the ramp and non-ramp events), a Bayesian optimization-based

approach is developed to optimize the parameters of DSPOT to enhance the quality of the training datasets and the corresponding performance of the NNs. *Ultimately, the proposed DSPOT-enhanced self-evolving neural networks (see Figure 1) form a closed loop for optimizing the performance of the wind generation forecast purely based on the data.*

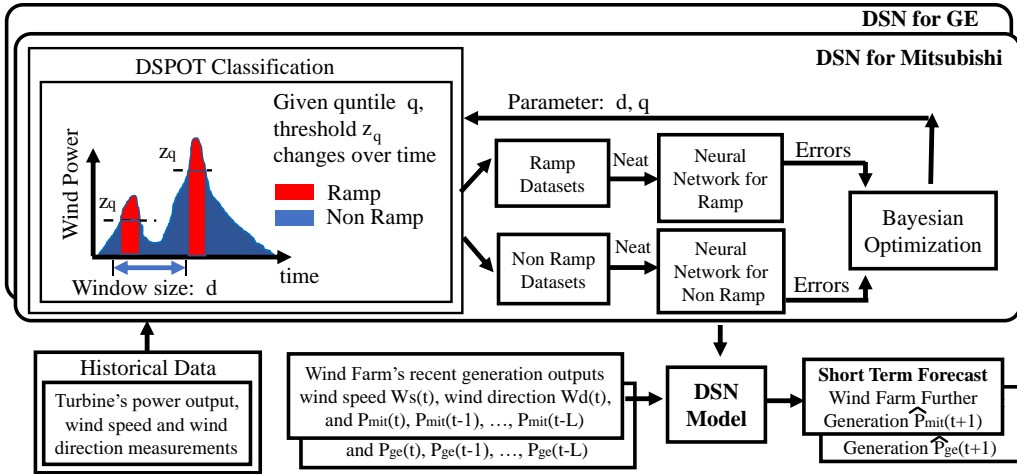

**Figure 1.** Illustration of the DSPOT-enhanced self-evolving neural networks.

Real-world wind farm generation measurements often exhibit distinct features, such as the non-stationarity and the heterogeneous dynamics for ramp and non-ramp events across different classes of wind turbines. Employing existing machine learning approaches without considering these features means wind farm generation cannot be accurately forecast, especially for ramp events. The contributions of this paper can be summarized as follows:

- We propose a Drifting Streaming Peaks-over-Threshold (DSPOT)-enhanced self-evolving neural networks-based short-term wind farm generation forecast, which is adaptive machine learning for wind farm generation forecasting. The proposed framework addresses the challenges of the non-stationarity and the ramp dynamics of wind farm generation and can greatly facilitate the integration of wind generation in the real world.
- The proposed method first classifies the wind farm generation data into the ramp and non-ramp datasets, where time-varying dynamics are captured by utilizing an adaptive thresholding framework to separate the ramp and non-ramp events, based on which different neural networks are trained to learn the dynamics of wind farm generation.
- As the efficacy of the neural networks relies on the quality of the training datasets (i.e., the classification accuracy of the ramp and non-ramp events), a Bayesian optimization-based approach is developed to optimize the parameters of the DSPOT algorithm to enhance the quality of the training datasets and the corresponding performance of the neural networks, which enables the model parameters to be adjusted automatically.
- The experimental results show that compared with other forecast approaches, the proposed forecast approach can substantially improve the forecast accuracy, especially for ramp events.

The remaining parts of this paper are organized as follows. Section 2 elaborates the distinct features of wind farm generation. Section 3 introduces the proposed wind farm generation forecast approach. Section 4 validates the performance of the proposed approach by using the real wind farm generation data. Section 5 summarizes the paper.

## 2. Data Description and Key Observations

This paper uses the same real wind generation data from a large wind farm as our previous works [36–38,40]. The wind farm has a rated capacity of 300.5 MW, where two classes of wind turbines are installed: Mitsubishi and GE turbines. There are 221 Mitsubishi turbines with a rated capacity of 1MW and 53 GE turbines with a rated capacity of 1.5 MW (see in Figure 2).

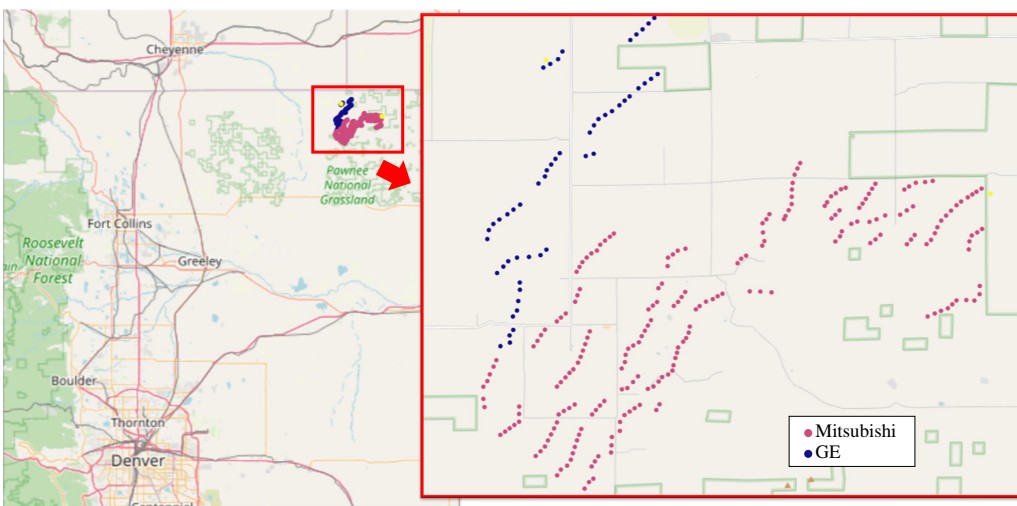

**Figure 2.** Locational distribution of the Mitsubishi and GE wind turbines in the wind farm.

Each class of wind turbines has distinct power curves as well as a cut-in and cut-off speed. For each class, a meteorological tower (MET), collocated with a wind turbine, is deployed to collect weather information. The instantaneous power outputs of each turbine together with the weather information are saved every 10 min for the years 2009 and 2010. In this paper, we use the power outputs of the Mitsubishi turbines $P_{mit}(t)$ and GE turbines $P_{ge}(t)$, the wind speed $W_s(t)$, and the wind direction $W_{dir}(t)$ to develop the proposed NNs.

From the measurements of the power outputs, we find (1) the non-stationarity of the power measurements and (2) the heterogeneous dynamics of the wind non-ramp and ramp events across each class of turbines as illustrated in Figure 3, where the cumulative distribution functions (CDFs) of the wind power measurements of two classes of turbines over different seasons of a year and different ramp events are presented.

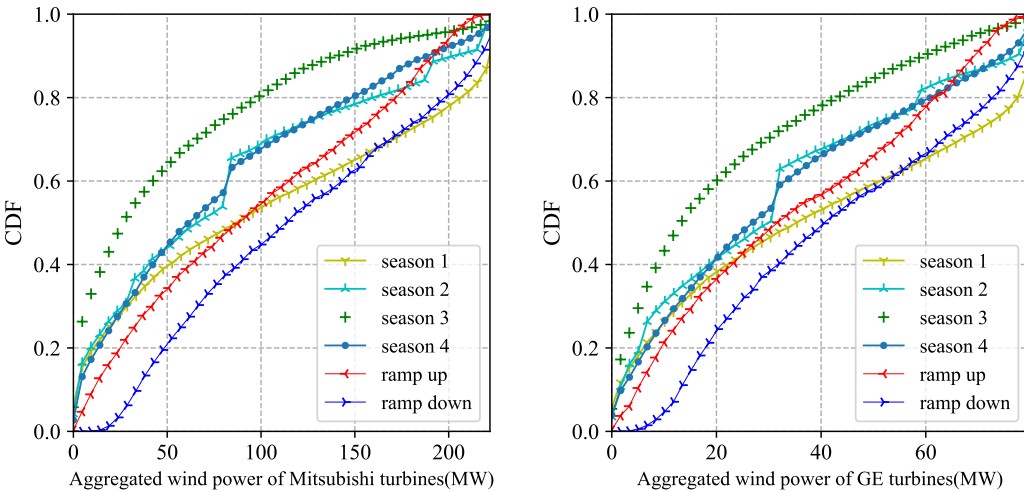

**Figure 3.** Empirical distribution of power outputs of GE and Mitsubishi turbines in 4 seasons and ramp events, where season 4 is from October to December.

In addition, it is shown in Figure 4 that the distributions of the ramps in different time windows *l* and different time periods are different and follow the generalized Pareto distribution (GPD).

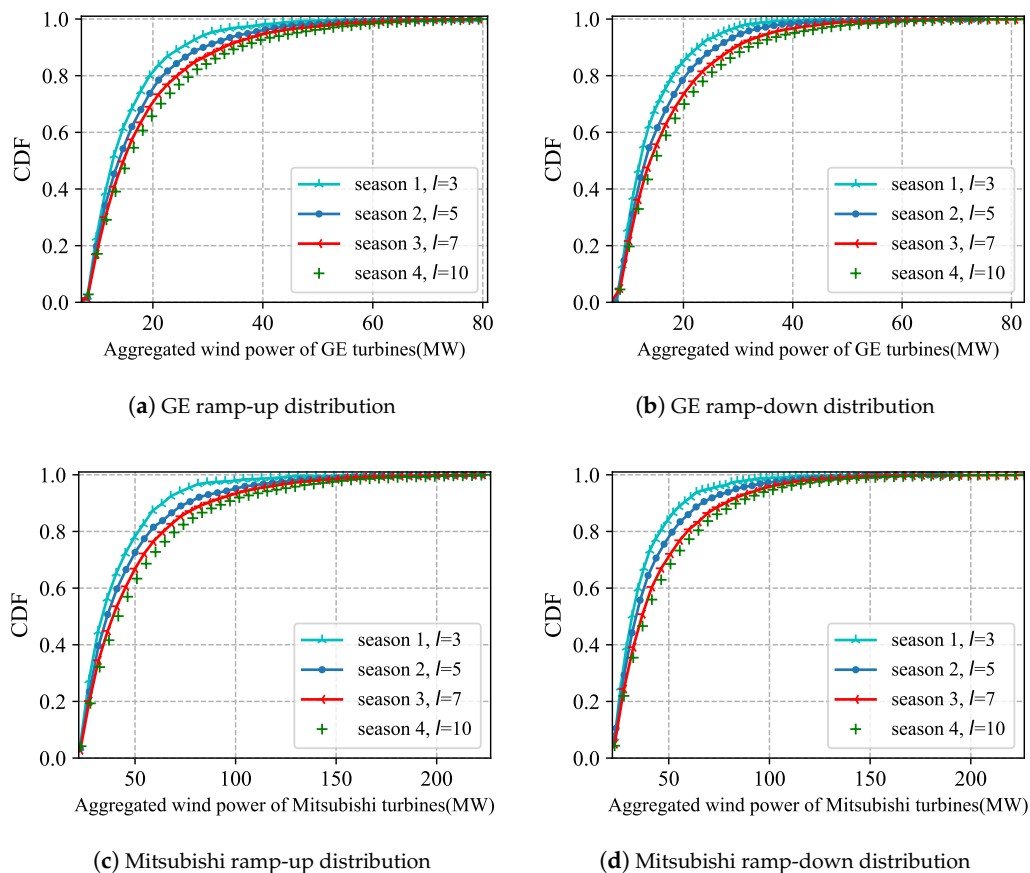

(**a**) GE ramp-up distribution

(**b**) GE ramp-down distribution

(**c**) Mitsubishi ramp-up distribution

(**d**) Mitsubishi ramp-down distribution

**Figure 4.** Empirical ramp distributions of GE and Mitsubishi turbines in different time windows *l* and different time periods, which follow the generalized Pareto distribution.

In the previous work [38], the non-stationarity is considered by developing seasonal self-evolving neural networks, where the ramp events are defined using fixed thresholds. As observed from Figure 4, fixed thresholds cannot fully capture the dynamics of wind ramp events. To address this challenge, we redefine the ramps by using dynamic thresholds, which change over time based on the dynamics of the ramp events, in order to reduce the forecast error of the wind farm generation, especially for ramp events.

## 3. DSPOT-Enhanced Self-Evolving Neural Networks

Motivated by the observations in Section 2, we seek to design a short-term forecast of a wind farm generation method that accounts for not only the heterogeneous dynamics of each class of wind turbines but also the time-varying dynamics of ramp and non-ramp events. Inspired by the success of artificial intelligence (AI) in a wide range of fields, our goal is to use neural networks (NNs) to learn these different dynamics of power outputs. Although there are several attempts along this line (e.g., ANNs [41] and LSTM [42]), these approaches use a single model and overlook the extreme ramp events, which leads to a poor forecast performance, especially for ramp events. Additionally, to train good NNs, it is critical to have high-quality training datasets (i.e., the ramp and non-ramp datasets should be well separated), which is a challenging task due to the time-varying dynamics of ramp and non-ramp events. Further, when training NNs, it is challenging to find the optimal topology as well as the hyperparameters of NNs.

To tackle these challenges, we propose DSPOT-enhanced self-evolving neural networks, namely the DSN, for the short-term wind farm generation forecast. The idea is to (1) first classify non-ramp and ramp events using DSPOT, which uses dynamic ramp thresholds to account for the time-varying dynamics of non-ramp and ramp events, and (2) then train different NNs for each dataset to learn the heterogeneous generation dynamics of the different classes of wind turbines, where these NNs can self-evolve based on the data, in order to account for the non-stationarity and reduce the overhead of tuning the topology and hyperparameters of NNs.

The design of our model is illustrated in Figure 1. The historical data are first classified into non-ramp, ramp-up, and ramp-down datasets by DSPOT, in which dynamic thresholds are determined based on recent observations in a moving window with size $d$, in order to appropriately define ramp and non-ramp events over time.

Then, we use NeuroEvolution of Augmenting Topologies [39] to train NNs using the classified datasets, in which the NNs evolve based on a genetic algorithm to obtain the best topology and hyperparameters of NNs. As a result, 6 NNs, i.e., 3 for Mitsubishi and 3 for GE, are built (see Figure 1). As the efficacy of NNs relies on the quality of training datasets, i.e., how good different ramp events are labeled, a Bayesian optimization-based method is proposed to optimize the parameters of DSPOT to enhance the quality of the training datasets and the corresponding performance of the NNs. Ultimately, the proposed DSPOT-enhanced self-evolving neural networks form a closed loop for optimizing the performance of wind farm generation forecast purely based on the data. In what follows, the design of each component of the model is described in detail.

### 3.1. DSPOT-Based Ramp Classifier

Based on extreme value theory, it is likely that extreme events follow a generalized Pareto distribution (GPD) [43], which is observed in wind power ramps in Figure 4. Thus motivated, we will develop a data-fitting technique using the GPD model to determine the dynamic threshold $z_{q^{cat}}(t)$ for different ramp events, where the index $cat \in \{up, down\}$ denotes the category of ramp events and $q^{cat}$ is the quantile of the corresponding ramp event distribution used to determine the threshold $z_{q^{cat}}(t)$. The idea is to first estimate the parameters of the GPD and then use the estimated GPD to find $z_{q^{cat}}(t)$ based on the quantile $q^{cat}$. To account for the time-varying dynamics of ramp events, the parameters of the GPD will be updated using the recent observed wind power in a moving window with size $d$.

Specifically, let $P_{class}(t)$ denote the wind power output at time $t$, where the index $class \in \{GE, Mitsubishi\}$ represents the class of wind turbines. In a specified time period $l$, ramp-up and ramp-down events can be separately expressed as:

$$
\begin{aligned}
P_{class}(t) - P_{class}(t-l) = \Delta P_{class}^l(t) > z_{q^{up}}(t), \\
P_{class}(t) - P_{class}(t-l) = \Delta P_{class}^l(t) < -z_{q^{down}}(t),
\end{aligned}
\tag{1}
$$

where $l$ and $q^{cat}$ are parameters to be tuned by BO (see Section 3.3) to determine the ramp events.

Based on the above definitions of ramp events, we classify the original dataset into ramp-up, ramp-down, and non-ramp datasets, i.e., 3 different datasets for each class of wind turbine. Let $\mathcal{X}_i^{class}, i \in \{up, down, non\}$ denote these 3 datasets, where $\mathcal{X}_{up}^{class}$ denotes the ramp-up dataset, $\mathcal{X}_{down}^{class}$ the ramp-down dataset, and $\mathcal{X}_{non}^{class}$ the non-ramp dataset. These datasets will be used to train NNs in Section 3.2. Clearly, the quality of these datasets (i.e., how well different ramp events can be separated) depends on the values of $z_{q^{up}}(t)$ and $z_{q^{down}}(t)$. In this section, we determine $z_{q^{up}}(t)$ and $z_{q^{down}}(t)$ using the GPD model. For ease of presentation, we present how to calculate the dynamic threshold $z_q(t)$ for ramp-up events by omitting the index $cat$ in the following. Correspondingly, the dynamic threshold for ramp-down events can be determined using the same procedure.

### 3.1.1. Calculating $z_q(t)$

We derive the log-likelihood of the GPD using the recent observations $\{\Delta P^l_{class}(t)\}_d$ in a moving window with size $d$:

$$
\begin{aligned}
L(\gamma, \xi; \{\Delta P^l_{class}(t)\}_d) &= -d \log \xi \\
&+ (\tfrac{1}{\gamma} - 1) \sum_{i=t-d+1}^{t} \log(1 - \tfrac{\gamma \Delta P^l_{class}(i)}{\xi}),
\end{aligned}
\tag{2}
$$

where $\gamma$ and $\xi$ are the parameters of the GPD ($\gamma \neq 0$). To estimate the parameters of the GPD, we find a solution $(\gamma^*, \xi^*)$ of $L$ by solving the following two equations:

$$
\frac{\partial L(\gamma, \xi; \{\Delta P^l_{class}(t)\}_d)}{\partial \gamma} = 0,
\tag{3}
$$

$$
\frac{\partial L(\gamma, \xi; \{\Delta P^l_{class}(t)\}_d)}{\partial \xi} = 0.
\tag{4}
$$

Grimshaw [43] has shown that if a solution $(\gamma^*, \xi^*)$ is obtained in this equation, the argument $\beta^* = \gamma^*/\xi^*$ is the solution to the scalar equation $u(\beta)v(\beta) = 1$, where

$$
u(\beta) = \frac{1}{|\mathcal{Y}_q|} \sum_{i=1}^{|\mathcal{Y}_q|} \frac{1}{1 + \beta Y_i},
\tag{5}
$$

$$
v(\beta) = 1 + \frac{1}{|\mathcal{Y}_q|} \sum_{i=1}^{|\mathcal{Y}_q|} \log(1 + \beta Y_i).
\tag{6}
$$

Here, a set $\mathcal{Y}_q = \{Y_i\}$ is defined for a given quantile $q$, i.e., $\mathrm{Prob}(\Delta P^l_{class}(i) > P^{th}_q) = q$, where $P^{th}_q > 0$ is the threshold associated with the quantile $q$. $\mathcal{Y}_q$ contains all $\Delta P^l_{class}(i)$ larger than $P^{th}_q$ with $Y_i = \Delta P^l_{class}(i) - P^{th}_q > 0$. $|\mathcal{Y}_q|$ denotes the cardinality of $\mathcal{Y}_q$. Based on Grimshaw trick [43], $\xi^*$ and $\gamma^*$ can be obtained using $\beta^*$ by

$$
\gamma^* = v(\beta^*) - 1,
\tag{7}
$$

$$
\xi^* = \gamma^*/\beta^*.
\tag{8}
$$

As there are multiple possible solutions of $\beta^*$, we need to find all the solutions in order to best estimate the GPD parameters $(\gamma, \xi)$ to fit the distribution of ramp events. It is noted that $1 + \beta Y_i$ must be strictly positive. As $Y_i$ is positive, we have $\beta^* \in (-\frac{1}{Y_{max}}, +\infty)$. Grimshaw also shows an upper-bound $\beta^*_{max}$:

$$
\beta^*_{max} = 2 \frac{\bar{Y} - Y_{min}}{(Y_{min})^2},
\tag{9}
$$

where $\bar{Y}$, $Y_{max}$, and $Y_{min}$ are the average amount, the maximum amount, and the minimum amount of $\mathcal{Y}_q$, respectively. Therefore, we can perform a numerical root search and find all possible solutions in $(-\frac{1}{Y_{max}}, \beta^*_{max})$, in which we choose the solution that maximizes the likelihood $L$.

Based on the estimated GPD, we can calculate $z_q(t)$ by solving the probability: $\mathrm{Prob}(\Delta P^l_{class}(i) > z_q(t))$. Based on [44], we leverage the probability of the exceedances of $\Delta P^l_{class}(i)$ over the threshold $P^{th}_q$,

$$
\begin{aligned}
&\mathrm{Prob}\{\Delta P^l_{class}(i) > z_q(t) | \Delta P^l_{class}(i) > P^{th}_q\} \\
&= (1 + \hat{\gamma}(\tfrac{z_q(t) - P^{th}_q}{\hat{\xi}}))^{-\frac{1}{\hat{\gamma}}}.
\end{aligned}
\tag{10}
$$

As $\text{Prob}(\Delta P^l_{class}(i) > P^{th}_q) = q$, we can solve

$$\text{Prob}(\Delta P^l_{class}(i) > z_q(t)) = q(1 + \hat{\gamma}(\frac{z_q(t) - P^{th}_q}{\hat{\xi}}))^{-\frac{1}{\hat{\gamma}}} \tag{11}$$

based on Bayesian theorem. Using (11), we can obtain $z_q(t)$ by

$$z_q(t) = P^{th}_q + \frac{\hat{\xi}}{\hat{\gamma}}\left(\left(\frac{q \cdot d}{|\mathcal{Y}_q|}\right)^{-\hat{\gamma}} - 1\right). \tag{12}$$

3.1.2. DSPOT Algorithm

Given a quantile $q^{cat}$, the DSPOT algorithm determines the dynamic threshold $z_{q^{cat}}(t)$ using the recent observations. Based on $z_{q^{cat}}(t)$, wind generation difference $\Delta P^l_{class}(t)$ will be labeled into ramp-up, ramp-down or non-ramp events, and the wind power of recent measurement $P_{class}(t)$ will be added into the corresponding dataset $\mathcal{X}^{class}_i$. The details of the DSPOT algorithm are provided in Algorithm 1.

Specifically, Algorithm 1 will first initialize the thresholds $z_{q^{up}}(t)$ and $z_{q^{down}}(t)$ using the first $d + l$ wind power measurements. Then, Algorithm 1 will update $z_{q^{up}}(t)$ and $z_{q^{down}}(t)$ using the new wind power measurement in the moving window with size $d$ in an online manner, based on which the new wind power measurement will be added into the corresponding dataset $X^{class}_i$. Algorithm 1 will be run for wind power measurements of each class of wind turbines.

---

**Algorithm 1** DSPOT

---

**Input:** $\{P_{class}(t)\}$, $d$, $l$, $q^{up}$, and $q^{down}$.
**Output:** $X^{class}_{up}$, $X^{class}_{down}$, and $X^{class}_{non}$.
**Initialization:**
(1) Calculate initial thresholds $z_{q^{up}}$, $z_{q^{down}}$ based on Section 3.1.1 using $\{P_{class}(t)|t = 1, \ldots, d + l\}$.
(2) Initialize $X^{class}_{up}$, $X^{class}_{down}$, and $X^{class}_{non}$ based on $z_{q^{up}}$ and $z_{q^{down}}$.
**End Initialization**
**For every** $t > d + l$ **in** $\{P_{class}(t)\}$
(1) Update $z_{q^{up}}(t)$ and $z_{q^{down}}(t)$ based on Section 3.1.1 using the recent observations $\{\Delta P^l_{class}(t)\}_d$.
(2) Classify $\Delta P^l_{class}(t)$ based on $z_{q^{up}}(t)$ and $z_{q^{down}}(t)$, and add $P_{class}(t)$ into the corresponding dataset $\mathcal{X}^{class}_i$.

---

*3.2. Self-Evolving Neural Network*

A self-evolving neural network (SEN) will be built for each dataset, $X^{class}_{up}$, $X^{class}_{down}$, and $X^{class}_{non}$. When training the neural networks (NNs), each element $P_{class}(t + 1)$ in $X^{class}_i$ is treated as the label and the corresponding features contain the wind speed $W_s(t)$, the change in wind direction degree $W_{dir}(t)$, and current power measurements $\{P_{class}(t), P_{class}(t - 1), \ldots, P_{class}(t - Lag)\}$, where $Lag$ depends on the measurements (see the discussion in Section 4.1). As demonstrated in Figure 5, NEAT [39] is used to train an NN. NEAT leverages a genetic algorithm (GA) to evolve the NN. It obtains the best network topology and the best weighting parameters by minimizing the forecast error, i.e., $\min \sum_t (\hat{P}_{class}(t) - P_{class}(t))^2$, where $\hat{P}_{class}(t)$ denotes the forecast from the NN.

As demonstrated in Figure 5, the workflow of NEAT contains random population generation, crossover, mutation, speciation, and evaluation by the fitness function. In this paper, the fitness function is defined using the forecast accuracy:

$$Fit = -\sum_t (\hat{P}_{class}(t) - P_{class}(t))^2. \tag{13}$$

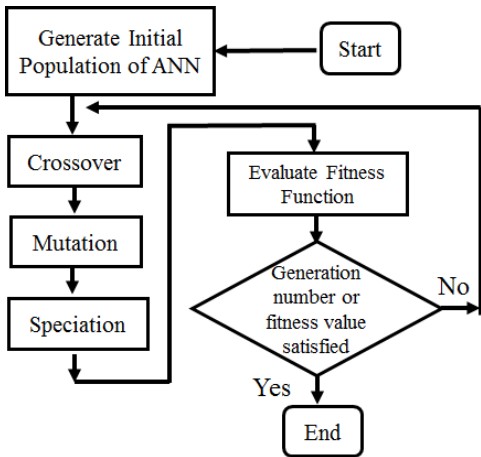

**Figure 5.** Workflow of NEAT.

Each gene in the population set corresponds to a neural network. We aim to find the best gene with the largest fitness value (i.e., the lowest prediction error). In NEAT, the topology of an NN is directly encoded into the gene by a direct encoding scheme [45] in order to avoid Permutations Problem [46] and Competing Conventions Problem [47]. Specifically, connection and node (list of inputs, hidden nodes, and outputs) are encoded. Every unit of connection gene describes the connection weight (W), output node (O), input node (I), enable gate (E), and the number of innovation (N) that corresponds to a consecutive arrangement of new generated node. The workflow of NEAT will be elaborated in the following.

First, initial population (i.e., a set of genes) is generated randomly. Each gene represents an NN. Note that under this random generation, a neural network might contain no route from inputs to outputs, and we will remove these NNs from the initial population. For example, Figure 6 shows an NN containing 3 inputs ($P_{class}(t), W_s(t), W_{dir}(t)$) and 1 output ($\hat{P}_{class}(t+1)$), where in the first unit of connect gene, I:1 O:5 W:0.5 indicates connection from Node 1 to Node 5 with weight of 0.5, and E:1 means that this is an enabled connection.

Node Gene:

| Node 1:<br>Wdir(t) | Node 2:<br>Ws(t) | Node 3:<br>$P_{class}(t)$ | Node 4:<br>Hidden | Node 5:<br>$\hat{P}_{class}(t+1)$ |
|---|---|---|---|---|

Connect Gene:

| I: 1<br>O:5<br>W:0.5<br>E: 1<br>N:1 | I: 2<br>O:5<br>W:0.7<br>E: 0<br>N:2 | I: 3<br>O:5<br>W:0.2<br>E: 1<br>N:3 | I: 4<br>O:5<br>W:0.4<br>E: 1<br>N:4 | I: 2<br>O:4<br>W:0.1<br>E: 1<br>N:5 | I: 3<br>O:4<br>W:0.5<br>E: 1<br>N:6 | I: 5<br>O:4<br>W:0.6<br>E: 1<br>N:11 |
|---|---|---|---|---|---|---|

**Figure 6.** Encoding of an NN with 1 output and 3 inputs.

After generating the initial population, NEAT iteratively optimizes the topology and connection weights of NNs using crossover and mutation. Specifically, nodes and connections of NNs are inserted or removed randomly based on the Poisson distribution [39]. For example, Figures 7 and 8 show possible mutations by appending a connection and a node to a neural network, respectively. After crossover and mutation, topologically homogeneous genes are classified as one speciation determined by compatibility distance [39].

Then, the fitness of species will be evaluated. If the highest fitness of species does not increase or the number of generations is achieved, NEAT will output the species with high fitness value, which will be used for wind generation forecast.

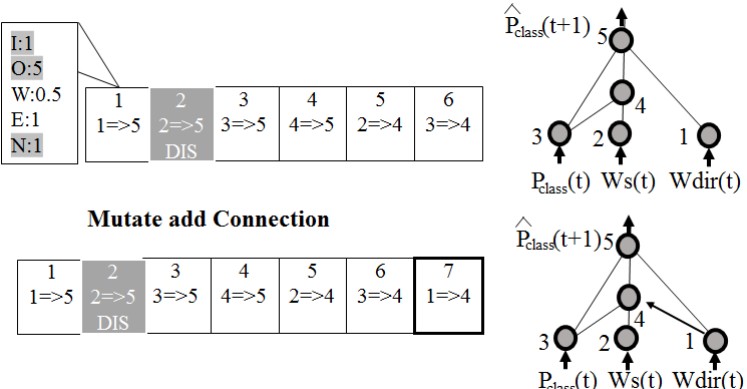

**Figure 7.** Mutation by appending a connection, where the link from Node 1 to Node 4 is inserted.

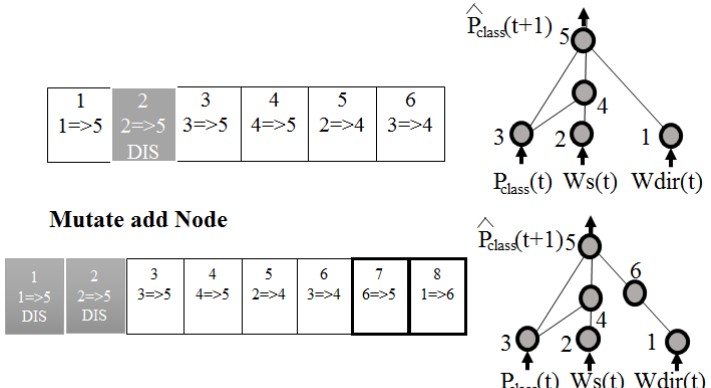

**Figure 8.** Mutation by appending a node, where Node 6 is inserted between Node 1 and Node 5.

### 3.3. Bayesian Optimization-Based Parameter Search

The performance of NNs depends on the quality of datasets $X_{up}^{class}$, $X_{down}^{class}$, and $X_{non}^{class}$ obtained by the DSPOT-based ramp classifier in Section 3.1. As the performance of the DSPOT-based ramp classifier relies on the parameters $\mathbf{b} = (l, d, q^{up}, q^{down})$, we develop a Bayesian optimization-based approach that can efficiently find the best parameters $\mathbf{b}^*$. The idea is to model the unknown function between the parameters and the training errors as a multivariate Gaussian distribution, and then use a computationally cheap acquisition function to guide the search for the best parameters.

Specifically, we introduce an acquisition function $\zeta(\cdot)$ as the optimization objective, which characterizes the expected training error improvement under $\mathbf{b}$,

$$\zeta(\mathbf{b}) = \mathbb{E}\left[ \sum_{class} \sum_{i} (F_i^{class}(\mathbf{b}^*) - F_i^{class}(\mathbf{b}))^+ \right], \tag{14}$$

where $F_i^{class}(\mathbf{b})$ denotes the training error of the NN trained under $\mathbf{b}$ using $X_i^{class}$ described in Section 3.2, and $F_i^{class}(\mathbf{b}^*)$ is the lowest error that has been obtained so far. It is assumed that the training errors $\{F_i^{class}(\mathbf{b}) | i \in \{up, down, non\}, class \in \{GE, Mitsubishi\}\}$ are random variables following the multivariate Gaussian distribution $\mathcal{G} \sim \mathcal{N}(m(\mathbf{b}), \Sigma(\mathbf{b}))$ with mean $m(\mathbf{b})$ and covariance $\Sigma(\mathbf{b})$. In each attempt, we find $\mathbf{b}$ that maximizes the acquisition function $\zeta(\mathbf{b})$. Then, we use this $\mathbf{b}$ as the input of Algorithm 1 to determine $X_{up}^{class}$, $X_{down}^{class}$, and $X_{non}^{class}$, based on which we evolve the NNs. Then, $\{F_i^{class}(\mathbf{b}) | i \in \{up, down, non\}, class \in \{GE, Mitsubishi\}\}$ will be added into a sample set $\mathcal{S}$, and the mean $m(\mathbf{b})$ and covariance $\Sigma(\mathbf{b})$ of $\mathcal{G}$ will be updated based on Bayesian optimization [48]. The details of the Bayesian optimization-based parameter search are given in Algorithm 2.

---

**Algorithm 2** Bayesian optimization-based parameter search

---

**Initialization:** Initialize $\mathcal{S} = \{(\mathbf{b}, \{F_i^{class}(\mathbf{b})\})\}$.
**For each attempt:**
(1) Find the parameter vector $\hat{\mathbf{b}}$ that maximizes $\zeta$, i.e., $\hat{\mathbf{b}} = \arg\max_{(\mathbf{b},\{F_i^{class}(\mathbf{b})\})\in\mathcal{S}} \zeta(\mathbf{b})$.
(2) Generate $X_{up}^{class}$, $X_{down}^{class}$, and $X_{non}^{class}$ based on Algorithm 1 using $\hat{\mathbf{b}}$, and evolve the NNs accordingly.
(3) Add the current training errors $\{F_i^{class}(\hat{\mathbf{b}})\}$ into the sample set $\mathcal{S} = \mathcal{S} \cup (\hat{\mathbf{b}}, \{F_i^{class}(\hat{\mathbf{b}})\})$, and update the parameters of $m(\mathbf{b})$ and $\Sigma(\mathbf{b})$ using $\mathcal{S}$.

---

*3.4. Short-Term Wind Farm Generation Forecast*

The proposed DSPOT-enhanced self-evolving neural networks (DSN) will train multiple NNs, which capture different dynamics of wind farm generation. When forecasting wind farm generation, we will first leverage the DSPOT-based ramp classifier to determine whether the current state of wind farm generation is in ramp up, ramp down, or non-ramp. Based on the classified state, we choose the corresponding NNs to forecast the wind farm generation.

Specifically, let the function $H_{\theta_i}^{class}(\cdot)$ represent the neural network with parameters $\theta_i$ (i.e., the best gene) trained using the datasets: $X_i^{class}(t) = \{W_s(t), W_{dir}(t), P_{class}(t), P_{class}(t-1), \ldots, P_{class}(t-Lag)\}$, the output of the neural network is

$$\hat{P}_{class}(t+1) = H_{\theta_i}^{class}(X_i^{class}(t)). \tag{15}$$

Based on the results of the ramp classifier, we pick the corresponding NNs (i.e., the best gene) for each class of wind turbines. Therefore, the wind farm generation forecast $\hat{P}_{ag}(t+1)$ can be achieved by:

$$\hat{P}_{ag}(t+1) = \hat{P}_{mit}(t+1) + \hat{P}_{ge}(t+1). \tag{16}$$

Equation (16) is the point forecast of wind farm generation.

Distributional forecasts are often needed to manage the uncertainty [49]. To this end, we leverage the collection of genes generated in NEAT and use the forecasts by these genes to develop distributional forecasts. Let $\{\hat{P}_{ag}^{(j)}(t)\}$ represent the set of forecasts offered by each gene $j$. It is assumed that the forecast error of the point forecasts follows the standard normal distribution with the mean $\mu_t$ and the variance $\sigma_t^2$ as follows:

$$\mu_t = \frac{1}{J}\sum_{j=1}^{J}\hat{P}_{ag}^{(j)}(t), \tag{17}$$

$$\sigma_t^2 = \frac{1}{J}\sum_{j=1}^{J}(\hat{P}_{ag}^{(j)}(t) - \mu_t)^2, \tag{18}$$

where $J$ is the number of genes. Under such assumption, we calculate the $(1-\alpha)$ confidence interval of the point forecasts (16) as follows:

$$[\hat{P}_{ag}(t+1) - Z(1-\frac{\alpha}{2})\sigma_{t+1}, \hat{P}_{ag}(t+1) + Z(1-\frac{\alpha}{2})\sigma_{t+1}], \tag{19}$$

where $Z(1-\frac{\alpha}{2})$ represents the point where the cumulative distribution function of the standard normal distribution is equivalent to $1-\frac{\alpha}{2}$.

**Remark 1.** *The proposed SENs can be trained offline. As the learning process of each SEN is based on different datasets, we can train these SENs on parallel. This can significantly reduce the training time of these SENs. Furthermore, the learning of SENs needs no AI experts to manually tune the topology and the hyperparameters; SENs can automatically adapt to the changing dynamics of*

*wind farm generation purely based on the data. This can greatly facilitate the implementation of the proposed method in reality.*

## 4. Case Studies of Real Wind Power Data

### 4.1. Experimental Setup

#### 4.1.1. Data

The data used in case studies are described in Section 2. Specifically, we use the data of year 2009 to train the proposed SENs and the data of year 2010 to validate the forecast performance of the proposed approach.

#### 4.1.2. Evaluation Metrics

Mean absolute error (MAE) and root mean square error (RMSE) are employed to evaluate the forecast performance, i.e.,

$$MAE = \frac{1}{N_t} \sum_t |\hat{P}_{ag}(t) - P_{ag}(t)|,$$

$$RMSE = \sqrt{\frac{1}{N_t} \sum_t |\hat{P}_{ag}(t) - P_{ag}(t)|^2}.$$

where $N_t$ is the number of data points in the test dataset.

#### 4.1.3. Parameter Tuning

As discussed in Section 3, the forecast performance of NNs greatly depends on the quality of training datasets, which hinges on the parameters $(l, d, q^{up}, q^{down})$ and $Lag$. To find the best $(l, d, q^{up}, q^{down})$, Algorithm 2 is run with 200 attempts.

To optimize $Lag$, we evaluate MAE under different values of $Lag$ (see Figure 9) and pick the one with the lowest MAE. It is observed that the lowest MAE is achieved when the feature dimension is 9 (i.e., $Lag = 7$).

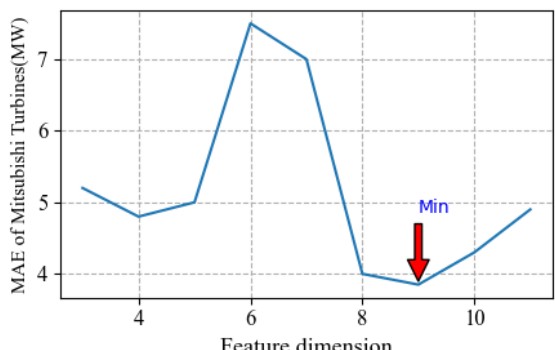

**Figure 9.** MAE versus feature dimension size.

#### 4.1.4. Benchmark

We compare the forecast performance of the proposed approach with the following benchmarks:

- The adaptive AR model [37];
- The Markov chain-based (MC) model [36];
- The SVM-enhanced Markov (SVM-MC) model [37];
- The seasonal NEAT (SNEAT) model trained by different season data without splitting ramp events;
- The NEAT model trained by the entire year data without splitting ramp events;
- The long short-term memory (LSTM) model trained by the entire year data;
- The artificial neural network (ANN);

- The seasonal self-evolving neural networks (SSEN) model [38].

The seasonal NEAT model considers four seasons, but it does not split ramp and non-ramp events in the training process, which would lead to a poor performance when ramp events occur. We use a prevailing structure of three layers to build the LSTM with the same configuration in [38]. The fully connected ANN is used, which includes three layers, and each layer contains 30 nodes.

*4.2. Experimental Results*

4.2.1. 10 min Ahead Forecast

In Tables 1 and 2, we compare the 10 min ahead forecast under different models for the whole year data and ramp events in the year 2010, respectively. The forecast results in terms of the MAE and RMSE are normalized using the nominal capacity of 300.5 MW of the wind farm.

**Table 1.** Forecast under different models over the whole year 2010.

| Error | AR | MC | SVM-MC | NEAT | SNEAT | LSTM | SSEN | ANN | DSN |
|---|---|---|---|---|---|---|---|---|---|
| MAE(%) | 2.441 | 2.413 | 2.214 | 1.734 | 1.778 | 1.799 | 1.704 | 1.826 | **1.661** |
| RMSE(%) | 3.974 | 3.524 | 3.342 | 3.030 | 3.074 | 3.072 | 3.023 | **2.993** | 2.996 |

**Table 2.** Forecast under different models over all ramps of the year 2010.

| Error | AR | MC | SVM-MC | NEAT | SNEAT | LSTM | SSEN | ANN | DSN |
|---|---|---|---|---|---|---|---|---|---|
| MAE(%) | 2.945 | 2.856 | 2.657 | 2.363 | 2.416 | 2.469 | 2.320 | 2.426 | **2.288** |
| RMSE(%) | 4.403 | 3.837 | 3.654 | 3.593 | 3.667 | 3.679 | 3.534 | 3.580 | **3.518** |

From Tables 1 and 2, we observe that the proposed approach (DSN) outperforms the benchmarks. Compared with the non-NN-based benchmarks (the AR, MC, and SVM-MC), the proposed approach improves the MAE at least 24.9% for the whole year data and at least 13.8% for the ramp events, respectively. Compared with the NN-based benchmarks, the improvement in the proposed approach (DSN) in terms of the MAE is at least 2.5% for the whole year and at least 1.3% for the ramp events. Such improvements are because of the splitting of the non-ramp and ramp events, which enables the DSN to more effectively learn the different dynamics of the GE and Mitsubishi turbines measurements under non-ramp and ramp events.

Figures 10–12 illustrate the prediction intervals for the three representative ramp events. The first chosen event is 5 January 2010 because there is a wind power ramp-up event from 4 a.m. to 5 a.m. with a ramp-up rate of 85 Megawatts per hour (MW/H). The second chosen event is 19 March 2010 because of the significant wind power fluctuation from 7 p.m. to 9 p.m. with both ramp-up and ramp-down events of an average ramp rate around 100 MW/H. The final chosen event is 9 October 2010 because of a remarkable ramp-down event from 3 a.m. to 5 a.m. with an average ramp rate of 66.5 MW/H. As demonstrated in those pictures, the actual wind farm generation is mostly confined in the prediction interval achieved from (19), regardless of the sharp ramps.

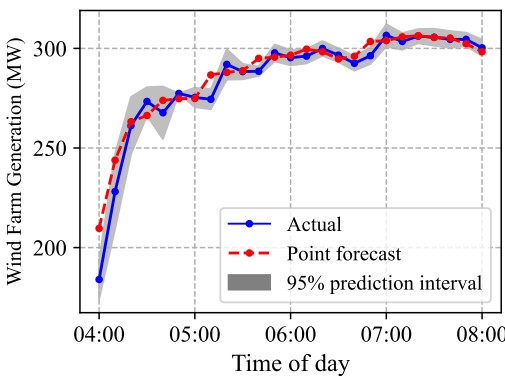

**Figure 10.** On 5 January 2010.

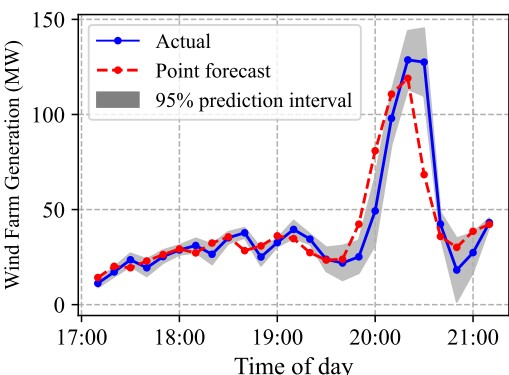

**Figure 11.** On 19 March 2010.

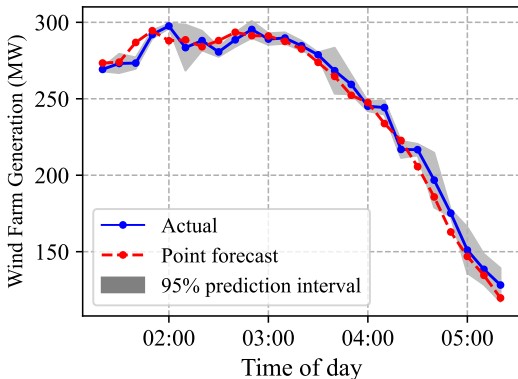

**Figure 12.** On 9 October 2010.

### 4.2.2. Other Forecasting Horizons

In Tables 3 and 4, we compare the forecast of different models under different horizons using the whole year data and ramp events in the year 2010, respectively. From Tables 3 and 4, we observe that the proposed approach outstrips the benchmarks under these forecasting horizons. It is observed in most cases that seasonal NEAT performs worse than NEAT (trained by using the entire year data). It is because the amount of data in a season is not enough for training a good NN compared to the entire year data.

**Table 3.** MAE of different models at different forecasting horizons over the whole year 2010.

| Model | 30 min | 40 min | 50 min | 60 min |
|-------|--------|--------|--------|--------|
| AR | 4.837 | 6.516 | 8.160 | 9.624 |
| MC | 4.733 | 6.233 | 7.551 | 8.727 |
| SVM-MC | 4.733 | 6.233 | 7.550 | 8.727 |
| NEAT | 4.804 | 5.851 | 6.939 | 7.640 |
| SNEAT | 5.064 | 6.322 | 7.681 | 8.277 |
| LSTM | 4.664 | 6.517 | 7.681 | 8.257 |
| SSEN | 4.852 | 5.970 | 7.095 | 7.862 |
| ANN | 4.755 | 5.846 | 6.580 | 7.491 |
| DSN | **3.755** | **4.220** | **4.746** | **5.069** |

For the 30 min ahead forecast, compared with the non-NN-based benchmarks (the AR, MC, and SVM-MC), the proposed approach improves the MAE at least 20.6% for the whole year data and at least 17.7% for the ramp events, respectively. Compared with the NN-based benchmarks (NEAT, SNEAT, LSTM, ANN, and SSEN), the enhancement of the proposed approach by the MAE is no less than 19.4% for the whole year data and at least 22.8% for the ramp events.

**Table 4.** MAE of different models at different forecasting horizons over ramp events of the year 2010.

| Model | 30 min | 40 min | 50 min | 60 min |
|-------|--------|--------|--------|--------|
| AR | 6.991 | 8.871 | 11.883 | 11.996 |
| MC | 6.592 | 8.426 | 10.654 | 11.091 |
| SVM-MC | 6.591 | 8.425 | 10.654 | 11.091 |
| NEAT | 7.255 | 8.379 | 10.366 | 10.274 |
| SNEAT | 7.427 | 8.612 | 10.471 | 10.385 |
| LSTM | 7.025 | 9.255 | 11.558 | 10.915 |
| SSEN | 7.092 | 8.182 | 9.727 | 9.849 |
| ANN | 7.109 | 8.087 | 9.384 | 9.627 |
| DSN | **5.420** | **5.595** | **7.023** | **6.197** |

For the 40 min ahead forecast, compared with the non-NN-based benchmarks (the AR, MC, and SVM-MC), the proposed approach improves the MAE at least 32.2% for the whole year data and at least 33.5% for the ramp events, respectively. Compared with the NN-based benchmarks (NEAT, SNEAT, LSTM, ANN, and SSEN), the enhancement of the proposed approach by the MAE is no less than 27.8% for the whole year data and at least 31.6% for the ramp events.

For the 50 min ahead forecast, compared with the non-NN-based benchmarks (the AR, MC, and SVM-MC), the proposed approach improves the MAE at least 37.1% for the whole year data and at least 34% for the ramp events, respectively. Compared with the NN-based benchmarks (NEAT, SNEAT, LSTM, ANN, and SSEN), the enhancement of the proposed approach by the MAE is no less than 31.6% for the whole year data and at least 27.7% for the ramp events.

For the 60 min ahead forecast, compared with the non-NN-based benchmarks (the AR, MC, and SVM-MC), the proposed approach improves the MAE at least 41.9% for the whole year data and at least 44.1% for the ramp events, respectively. Compared with the NN-based benchmarks (NEAT, SNEAT, LSTM, ANN, and SSEN), the enhancement of the proposed approach by the MAE is no less than 33.6% for the whole year data and at least 37% for the ramp events.

### 4.2.3. Distributional Forecast

The continuous rank probability score (CRPS) is used to evaluate the performance of the proposed distributional forecasts. The CRPS is defined as:

$$\text{CRPS} = \frac{1}{N_t} \sum_t \int_0^{P_{ag}^{max}} (\hat{F}_t(x) - U(x - P_{ag}(t))) dx \tag{20}$$

where $\hat{F}_t(x)$ is the cumulative density function (cdf) obtained by using the distributional forecast. In addition, $U(.)$ is a unit step function that equals to 1 if $x > P_{ag}(t)$ and 0 otherwise. Generally, the lower the CRPS, the more accurate the distributional forecast is. In Tables 5 and 6, we compare the forecast of different NN-based models under different horizons using the whole year data and ramp events in the year 2010, respectively. The results of the non-NN models can be found in [37]. We observe that our model performs much better than other benchmarks for longer prediction horizons (normally longer than 30 min) where the wind ramps are large, while the performance is similar for the 10 min forecast. This indicates the superior performance of the proposed method on handling the uncertainty of the wind.

For the 30 min ahead forecast, compared with the NN-based benchmarks (NEAT, SNEAT, LSTM, ANN, and SSEN), the proposed approach improves the CRPS at least 24.9% for the whole year data and at least 21.7% for the ramp events, respectively.

For the 40 min ahead forecast, compared with the NN-based benchmarks (NEAT, SNEAT, LSTM, ANN, and SSEN), the proposed approach improves the CRPS at least 26% for the whole year data and at least 30% for the ramp events, respectively.

For the 50 min ahead forecast, compared with the NN-based benchmarks (NEAT, SNEAT, LSTM, ANN, and SSEN), the proposed approach improves the CRPS at least 33.3% for the whole year data and at least 38.8% for the ramp events, respectively.

For the 60 min ahead forecast, compared with the NN-based benchmarks (NEAT, SNEAT, LSTM, ANN, and SSEN), the proposed approach improves the CRPS at least 35.8% for the whole year data and at least 35.2% for the ramp events, respectively.

**Table 5.** CRPS of different NN models distributional forecast (normalized by $P_{ag}^{max}$) over the year of 2010.

| Model | 10 min | 30 min | 40 min | 50 min | 60 min |
|-------|--------|--------|--------|--------|--------|
| NEAT | 1.628 | 4.595 | 5.761 | 6.451 | 7.371 |
| SNEAT | 1.601 | 4.655 | 5.773 | 6.751 | 7.512 |
| LSTM | 2.000 | 4.644 | 5.574 | 6.947 | 7.309 |
| SSEN | **1.584** | 4.614 | 5.776 | 6.704 | 7.464 |
| ANN | 1.651 | 4.84 | 5.844 | 6.756 | 7.755 |
| DSN | 1.611 | **3.598** | **4.120** | **4.303** | **4.725** |

**Table 6.** CRPS of different NN models distributional forecast (normalized by $P_{ag}^{max}$) of all ramps over the year of 2010.

| Model | 10 min | 30 min | 40 min | 50 min | 60 min |
|-------|--------|--------|--------|--------|--------|
| NEAT | 2.296 | 7.048 | 8.123 | 9.585 | 9.897 |
| SNEAT | 2.234 | 6.995 | 7.952 | 9.632 | 9.743 |
| LSTM | 2.719 | 6.944 | 7.781 | 9.799 | 9.290 |
| SSEN | **2.186** | 6.845 | 7.975 | 9.447 | 9.513 |
| ANN | 2.276 | 7.192 | 8.010 | 9.739 | 9.941 |
| DSN | 2.276 | **5.137** | **5.445** | **5.776** | **6.018** |

### 4.2.4. Model Updating

The training time for the self-evolving NN depends on the number of training samples. In our case, the number of ramp-up and ramp-down events in the training datasets is less than 4000, and updating the corresponding models takes only about 3–5 min using a machine with Dual-sockets Intel(R) Xeon(R) CPU E5-2630 v4 @ 2.20GHz. The updating time is much less than the forecasting horizons, and therefore our model can work well in practice.

#### 4.2.5. Discussions

Based on the experimental results, we observe that the NN-based models outperform the non-NN-based models. By breaking the training datasets into ramp and non-ramp training datasets for distinct classes of wind turbines, the performance of the NNs can be improved.

Further, the proposed DSPOT-based ramp classifier can better split the ramp and non-ramp events using dynamic thresholds and therefore better capture the heterogeneous dynamics of wind farm generation. Moreover, the proposed DSN can automatically adapt to the changing dynamics of wind farm generation over time, and the model updating time for the DSN is low. Specifically, the number of ramp-up and ramp-down events in the training datasets is less than 4000, and updating the corresponding models takes only about 3–5 min using a machine with Dual-sockets Intel(R) Xeon(R) CPU E5-2630 v4 @ 2.20GHz. The updating time is much less than the forecasting horizons, and therefore our model can work well in practice.

As shown in the experiments, the performance improvement for the point forecast and the distributional forecast is smaller in the 10 min horizon while the improvement is higher in the 60 min horizon. This might be due to the fact that some baseline models (e.g., the AR, SNEAT, LSTM, and ANN) are not considering ramp events which leads to a deeper forecast degeneration with a longer prediction horizon. Although the SSEN in our previous work [38] considered the ramp events, it leverages a fixed threshold to distinguish the ramp events. Because our proposed framework adjusts the ramp thresholds dynamically, the accuracy results are superior compared to the existing benchmarks.

### 5. Conclusions

We develop the DSPOT-enhanced self-evolving neural networks for the short-term wind power forecast. Specifically, the proposed approach initially classifies the wind farm generation data into ramp and non-ramp datasets using DSPOT, which leverages the dynamic ramp thresholds to account for the time-varying dynamics of the ramp and non-ramp events. We then train different NNs based on each dataset to learn the different dynamics of wind farm generation by NEAT, which are able to obtain the best network topology and weighting parameters. As the efficacy of the neural networks relies on the quality of the training datasets (i.e., the classification accuracy of the ramp and non-ramp events), a Bayesian optimization-based approach is developed to optimize the parameters of DSPOT to enhance the quality of the training datasets and the corresponding performance of the neural networks. The experimental results show that the proposed approach outperforms other forecast approaches.

In the future work, we plan to leverage the generative adversarial networks (GAN)-based models to better classify the ramp events which in turn would improve the quality of the training datasets.

**Author Contributions:** Conceptualization, Y.L. and L.Y.; methodology, Y.L.; coding, Y.L.; validation, Y.L.; formal analysis, Y.L. and A.G.; investigation, Y.L.; resources, L.Y.; data curation, Y.L. and L.Y.; writing—original draft preparation, Y.L. and L.Y.; writing—review and editing, Y.L. and A.G.; visualization, Y.L.; supervision, L.Y.; project administration, L.Y.; funding acquisition, L.Y. All authors have read and agreed to the published version of the manuscript.

**Funding:** This work is supported in part by the NSF under Grants IIS-1838024, CNS-1950485, and OIA-2148788.

**Data Availability Statement:** Not Applicable due to privacy restrictions, the study does not report any data.

**Conflicts of Interest:** The authors declare no conflict of interest.

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
