# Peer review of "Drifting Streaming Peaks-Over-Threshold-Enhanced Self-Evolving Neural Networks for Short-Term Wind Farm Generation Forecast"

_futureinternet, doi:10.3390/fi15010017_

Round 1

Reviewer 1 Report

Dear Authors.

This paper (Drifting Streaming Peaks-over-Threshold Enhanced Self-evolving Neural Networks for Short-term Wind Farm Generation Forecast) is interesting work, Short-term Wind Farm Generation Forecast.

It is very good paper.

However, I have some Minor comments.

Strengths:

1.  An interesting partition in Short-term Wind Farm Generation Forecast.

Weaknesses:

1. Reason behind deploying in a new Short-term Wind Farm Generation Forecast should be elaborated more.

2. More Background....

3. Contribution

3.1. What is Drifting Streaming Peaks-over-Threshold Enhanced Self-evolving Neural Networks ? (Contribution more)

3.2. What is main idea?

4. Moderate English changes required.

-Text errors and text editing.

5. The paper should be conducted proof-reading.

Author Response

This paper (Drifting Streaming Peaks-over-Threshold Enhanced Self-evolving Neural Networks for Short-term Wind Farm Generation Forecast) is interesting work, Short-term Wind Farm Generation Forecast. It is very good paper. However, I have some Minor comments. 

Point 1: Strengths: An interesting partition in Short-term Wind Farm Generation Forecast. Weaknesses: Reason behind deploying in a new Short-term Wind Farm Generation Forecast should be elaborated more.

 Response 1:  We thank the reviewer for the valuable comments.

The motivations for our paper are summarized as follows: 

1) “Although neural network (NN) based methods may improve the forecast accuracy to a certain degree, existing NN-based approaches have poor performance for the ramp events since they have different dynamics than non-ramp events. It has been shown that NNs perform poorly if extreme (or ramp) events are overlooked[1]. Employing NNs without considering these distinct features of wind farm generation cannot accurately forecast wind farm generation for ramp events.

2) In our previous works [2,3], self-evolving neural networks are built for different seasons, and ramps are defined using fixed thresholds. However, it is observed that the dynamics of wind ramps may change within each season. Therefore, it is challenging to use fixed thresholds to accurately capture the dynamics of wind ramps. To address this challenge, this paper proposes a dynamic threshold-based approach that can adapt to the time-varying dynamics of wind ramps. We argue that our proposed framework offers superior forecasting accuracy in presence of ramp events.

Point 2: More Background....

 Response 2: We thank the reviewer for the valuable comments.

We added more background discussion by adding more references to the manuscript.

Point 3: Contribution: What is Drifting Streaming Peaks-over-Threshold Enhanced Self-evolving Neural Networks ? (Contribution more) What is main idea?

 Response 3: We thank the reviewer for the valuable comments.

Drifting Streaming Peaks-over-Threshold (DSPOT) is a method that efficiently captures the time-varying dynamics of the ramp and non-ramp events during the training stage which in turn improves the wind farm generation forecasting accuracy significantly. Specifically, our proposed method first classifies the wind farm generation data into the ramp and non-ramp datasets by taking into account the time-varying dynamics using the DSPOT algorithm. In the next step, we leverage the NeuroEvolution of Augmenting Topologies (NEAT) framework to train one neural network for each dataset to learn different dynamics of wind farm generation.  Note that the neural network topology and weighting parameters will be obtained using the genetic algorithm in the NEAT framework. We also develop a Bayesian optimization-based approach to optimize the parameters of the DSPOT algorithm to enhance the quality of the training datasets as the efficacy of the neural networks relies on the quality of training datasets (i.e., the classification accuracy of the ramp and non-ramp events).

We would like to emphasize that the main idea of this paper was inspired by reviewing previous studies in wind generation forecasting [2-4]. Reviewing the literature revealed the following insights: 1) the non-stationarity of power measurements, and 2) the heterogeneous dynamics of wind non-ramp and ramp events across each class of turbine.  Without considering these distinct features, wind farm generation cannot accurately forecast wind farm generation, especially for ramp events. Our proposed method leverages different neural networks to learn different dynamics of wind generations. Moreover, we leverage the DSPOT algorithm to find the dynamic thresholds for defining ramps and non-ramps.

Point 4: Moderate English changes required. -Text errors and text editing.

Response 4: We thank the reviewer for the valuable comments.

We made some changes to remove the typos and improve the readability.

Point 5: The paper should be conducted proof-reading.

Response 5: We thank the reviewer for the valuable comments.

We proofread the manuscript to remove possible typos and improve the readability.

References:

[1] Ding, D.; Zhang, M.; Pan, X.; Yang, M.; He, X. Modeling extreme events in time series prediction. In Proceedings of the Proceedings of the 25th ACM SIGKDD International Conference on Knowledge Discovery & Data Mining, 2019, pp. 1114–1122.

[2] He, M.; Yang, L.; Zhang, J.; Vittal, V. A spatio-temporal analysis approach for short-term forecast of wind farm generation. IEEE Transactions on Power Systems 2014, 29, 1611–1622.

[3] Yang, L.; He, M.; Zhang, J.; Vittal, V. Support-vector-machine-enhanced markov model for short-term wind power forecast. IEEE Transactions on Sustainable Energy 2015, 6, 791–799.

[4] Liu, Y.; Ghasemkhani, A.; Yang, L.; Zhao, J.; Zhang, J.; Vittal, V. Seasonal Self-evolving Neural Networks Based Short-term Wind Farm Generation Forecast. In Proceedings of the 2020 IEEE International Conference on Communications, Control, and Computing Technologies for Smart Grids (SmartGridComm). IEEE, 2020, pp. 1–6.

Reviewer 2 Report

The article is interesting and describes an important topic.

Corrections and additions require:

- improved readability of figure 2

- expansion of literature research and the number of citations - 30 literature references is not enough

- a clear indication of the limitations and potential area for further possible research

- extension of the section with the discussion of the results and the section with the summary

Author Response

The article is interesting and describes an important topic. Corrections and additions require:

Point 1: improved readability of figure 2

Response 1:  We thank the reviewer for the valuable comments.

We replotted figure 2 with a high-resolution format.

Point 2:  expansion of literature research and the number of citations - 30 literature references is not enough.

Response 2:  We thank the reviewer for the valuable comments.

We expanded the literature review by adding more discussion and references.

Point 3:  a clear indication of the limitations and potential area for further possible research

Response 3:  We thank the reviewer for the valuable comments.

Improving the quality of ramp and non-ramp datasets can improve forecasting accuracy. The Generative Adversarial Network (GAN) is an interesting direction for pattern recognition, which can be potentially used in capturing the dynamics of ramp events. More research work can be done in this direction.

We added the following discussion in the conclusion section to outline future work. “In the future work, we plan to leverage the Generative Adversarial Networks(GAN) based models to better classify the ramp and non-ramp events which in turn can improve the quality of the  training datasets.”

Point 4:  extension of the section with the discussion of the results and the section with the summary

Response 4:  We thank the reviewer for the valuable comments.

We added the following paragraph to the discussion section (i.e., section 4.2.5) to analyze our findings. “As shown in the experiments, the performance improvement for the point forecast and the distributional forecast is smaller in the 10-minutes horizon while the improvement is higher in the 60-minutes horizon. This might be due to the fact that some baseline models (e.g., AR, SNEAT, LSTM, ANN)  are not considering ramp events which leads to deeper forecast degeneration with a longer prediction horizon. Although SSEN in our previous work [1] considered the ramp events, it leverages a fixed threshold to distinguish the ramp events. Since our proposed framework adjusts the ramp thresholds dynamically, the accuracy results are superior compared to the existing benchmarks.”

We added the following discussion in the conclusion section to outline future work. “In the future work, we plan to leverage the Generative Adversarial Networks(GAN) based models to better classify the ramp events which in turn would improve the quality of the  training datasets.”

References:

[1] Liu, Y.; Ghasemkhani, A.; Yang, L.; Zhao, J.; Zhang, J.; Vittal, V. Seasonal Self-evolving Neural Networks Based Short-term Wind Farm Generation Forecast. In Proceedings of the 2020 IEEE International Conference on Communications, Control, and Computing Technologies for Smart Grids (SmartGridComm). IEEE, 2020, pp. 1–6.

Reviewer 3 Report

 In this article, the authors focused on the determination of proper hyperparameters of developed model using a Bayesian Optimization based Parameter Search scheme, considering dynamics of wind turbines. The article is interesting and meaningful, and it has a clear aim. Some recommendations are given:

1.  Explain how to determine the initial parameters of the developed model in different climate conditions

2. Figures 1~3  and Figs 9~11 are very blurry,  please change them with high resolution

3.  How do you avoid the overfitting problem?

Author Response

In this article, the authors focused on the determination of proper hyperparameters of developed model using a Bayesian Optimization based Parameter Search scheme, considering dynamics of wind turbines. The article is interesting and meaningful, and it has a clear aim. Some recommendations are given:

Point 1: Explain how to determine the initial parameters of the developed model in different climate conditions

Response 1:  We thank the reviewer for the valuable comments.

We would like to emphasize that we leverage the Bayesian Optimization (BO) based approach to optimize the parameters of the DSPOT algorithm. Since we use a systematic parameter tuning framework (i.e., BO), model parameters will be adjusted automatically; hence, the model will not be affected by different climate conditions. The idea for BO is to use Bayes Theorem to explore the search space to find the optimal value for an acquisition function. In our case, given any initial random combination of the parameters (forms a parameter vector), our model will start evaluating the model by offering a training error. Based on such an error and the acquisition function, the  BO will decide what is the next parameter vector to evaluate for the next iteration. After several iterations, the best combination of those parameters will be determined after the training phase has finished. More details on the BO algorithm for parameter tuning can be found in [1].

Point 2: Figures 1~3 and Figs 9~11 are very blurry, please change them with high resolution

Response 2:  We thank the reviewer for the valuable comments.

We have replotted those figures and used high-resolution pictures.

Point 3: How do you avoid the overfitting problem?

Response 3:  We thank the reviewer for the valuable comments.

An overfitted model is identified by monitoring the performance of the model during the training stage by evaluating both training and validation datasets. If the training error keeps on decreasing while the validation error reaches the inflection point(concave up), it can be defined as overfitting.

In our training phase, we randomly split the training and validation sets, when the fitness value of the validation set drops monotonously for 10 generations, we stop the training. Thus, this can avoid the overfitting problem.

References:

[1] J. Snoek, H. Larochelle, and R. P. Adams, “Practical bayesian optimization of machine learning algorithms,” in Advances in Neural Information Processing Systems, vol. 25. Curran Associates, Inc., 2012.

Reviewer 4 Report

The paper presents a novel adaptive wind forecasting method using neural networks where different neural networks are trained for ramp up, ramp down and non ramp events. The architecture is further optimized using Bayesian approach and overall it is a comprehensive and original work presented within the area of research. 

I have no major concerns regarding the article only a few minor ones: 

1. State the contributions of the work clearly in the introduction section. 

2. Include some results in the abstract. 

3. Please insert a figure that shows the entire data that is used.

4. Future work regarding the proposed approach should be included in the conclusions.  

Round 2

Reviewer 2 Report

Thanks for the corrections and additions. I accept the article as it stands.